# OpenReview forum: "SPILLage: Agentic Oversharing on the Web"
_ICLR.cc/2026/Conference — Submitted to ICLR 2026_

### Official Review · Reviewer_wk34 · 2025-10-28

**Soundness:** 3
**Presentation:** 2
**Contribution:** 2
**Rating:** 2
**Confidence:** 4

**Summary:**

The paper introduces SPILLAGE, a framework for analyzing how web agents handle user resources and potentially disclose sensitive information to external parties. The authors identify four categories of oversharing: explicit content, implicit content, explicit behavioral, and implicit behavioral. They developed a benchmark to evaluate web agents on live e-commerce sites and found that agents tend to overshare more information than necessary, compromising user privacy. The study also discovered that reducing unnecessary information in user prompts can improve task success rates.

**Strengths:**

- Testing agent's oversharing in real-world web environment is certainly important. This real-world benchmark allows doing that (although I do have some concerns on this listed in the next section).
- Taxonomy of oversharing categories is quite interesting and it provides a nuanced understanding of the issue.

**Weaknesses:**

- I don't think the use of live websites is a good idea here. It makes it challenging to control variables and reproduce results, which is a crucial aspect of scientific experimentation. Whereas environments such as BrowserGym, WebArena allow this.
- By looking into examples, the definition and detection of implicit (both content and behavior) oversharing may be subjective, and it's unclear whether humans would perform better in similar situations. For example, for a repeated search of products in the range of $400-$600, what should AI do to avoid oversharing this? Maybe conducting a human study to compare human and agent performance could provide valuable insights.
- I guess the above concern is also relevant to explicit oversharing. By inspecting prompts in appendices, I noticed that models are not given any hint about the oversharing nor any instructions to preserve sensitive data. Since these models are primarily designed to be instruction-following, I'm wondering if this benchmark is asking too much from the model. For example, there is no mention that email chain in D.3 should be treated with some care and agent generally is not instructed to be sensitive-aware. So, it might be a fair game to include part of it in the product search?
- The evaluation methodology relies on LLM judges to detect oversharing events, which may not be perfect. There's a risk that the judges may miss certain types of oversharing or incorrectly classify benign behavior as oversharing. Was there any study to evaluate accuracy of LLM judge
- The paper doesn't thoroughly discuss straightforward solutions to mitigate oversharing, such as modifying the agent's design or implementing additional safety protocols (e.g. via system prompt).

**Questions:**

- example in lines 256-260 doesn't make sense to me. How iPhone is related to the "divorce" example?
- did authors use a separate 'tool' role (for GPT models) to interact with Browser? Or everything in the user-context?

---

> ### Author Response · Authors · 2025-11-27
>
> We sincerely appreciate Reviewer wk34’s thoughtful comments. Below, we provide our response to the concern raised.
>
> **Response to Weakness 1 and Question 1 (*live websites*)**: We agree that controlled environments like BrowserGym or WebArena can offer stronger reproducibility and more precise control over experimental variables. Our choice to evaluate on real, live websites was intentional: we wanted to observe oversharing as it occurs in realistic, dynamic settings, where interfaces change, recommendation systems evolve, and agent behavior may drift, which are the conditions that better reflect how these systems would operate in practice. While controlled benchmarks make it easier to guarantee consistent outputs across repeated runs, they do not fully capture the complexity or unpredictability of commercial platforms where real privacy risks occur. To address this trade-off, we plan to supplement our real-world evaluations with repeated trials of the same prompts to report variance and confidence intervals, ensuring that our findings are both realistic and statistically supported.
>
> **Response to Weakness 2 and 4 (*subjectivity and LLM-Judge concern*)**: We agree that relying on an LLM-based judge introduces limitations, especially for borderline or implicit cases where interpretation can be subjective. We chose this approach primarily because it aligns with how recent work in this space (e.g., AgentDAM) evaluates privacy leakage, and because it allows scalable step-level auditing across large numbers of trajectories. That said, we acknowledge that LLM-judging is not perfect, and improving detection accuracy is an open research challenge. As part of our future work, we are definitely interested in validating the LLM-judge against human annotations and exploring hybrid or rule-augmented approaches, particularly or implicit behavioral cases where inference depends on contextual oversharing.
>
> **Response to Weakness 3 (*asking too much to the model*)**: We understand the concern, and it’s a fair question. At first glance, it may seem like we’re asking the model to do something it was never instructed to do. However, modern web agents are marketed and deployed as *LLM-powered assistants*, where systems that should reason over context, adapt to user intent, and respect implicit norms without being explicitly told at every step. In this setting, we would like to emphasize that contextual integrity isn’t an extra requirement, but rather part of the main requirement. When users delegate tasks and grant access to personal resources (emails and chats), they reasonably expect the agent to distinguish what is necessary for the task from what is merely present in the context. In real deployment, the users can’t (and often won’t) manually sanitize every detail before letting the agent act. Because of that, we believe it is reasonable to expect agents to handle sensitive information with care, rather than surface everything available just because it exists in the prompt. So while we aren’t prompting the models to “please protect privacy”, it reflects how users will interact with these systems in practice. Our benchmark is meant to reveal whether current agents can meet that expectation.
>
> **Response to Weakness 5 (*mitigation*)**: We appreciate this concern, and we agree that one natural next step is to explore how far simple guardrails can mitigate oversharing. As mentioned as part of our response to reviewer fDzh, we ran a preliminary experiment where we modified the system prompt to explicitly instruct the agent to “be careful and avoid using task-irrelevant information.” Interestingly, rather than reducing leakage, this actually increased oversharing (86.0% vs. 58.9% in the regular setting). Our hypothesis is that drawing attention to sensitive attributes makes them more cognitively “available” to the model or anchors the agent’s reasoning around those details, resulting in more, not less, disclosure.

---

> ### Author Response · Authors · 2025-11-27
>
> **Response to Question 1 (*iPhone*)**: To avoid confusion, we have added a clearer example at the following link: https://anonymous.4open.science/r/SPILLage-D2AD/tasks/. In this prompt, the agent is asked to find “affordable glucose test strips on Amazon.com,” but it also has access to additional user resources such as email and chat history. These resources include unrelated personal details, such as discussions about purchasing an iPhone and messages about a recent divorce. Since neither of these details is required to complete the shopping task, any mention or use of them constitutes oversharing under our framework. We appreciate reviewer wk34 for raising this point, as it helped us refine the example and better communicate how task-irrelevant information is defined in our benchmark.
>
> **Response to Question 2 (*tool*)**: For our experiments, the user information was embedded directly in the prompt rather than retrieved through explicit tool calls. However, our setup reflects the realistic assumption that an agent with granted permissions could access the same personal resources during execution. To avoid ambiguity, we’ve shared the full dataset at the link above so reviewers can clearly see how the prompts are structured and how the information is presented to the agent.

---

### Official Review · Reviewer_S1Ni · 2025-11-01

**Soundness:** 2
**Presentation:** 3
**Contribution:** 2
**Rating:** 4
**Confidence:** 4

**Summary:**

This paper presents SPILLAGE, a framework for analyzing the oversharing behaviors of web agents. It categorizes the oversharing behaviors into four categories along two dimensions: directness (explicit vs. implicit) and mode (textual vs. non-textual actions). Based on the four categories, it crafts a benchmark consisting of prompts at three levels based on synthetic personas, and then evaluated two agentic frameworks: Browser-Use and AutoGen, along with three LLMs: o3, o4-mini, and gpt-4o. They found the oversharing behaviors were prevalent during the interactions of the agents with the real-world websites and removing inappropriate information doesn't reduce utility, showing a possible mitigation direction to improve both privacy and utility.

**Strengths:**

- The study on contextual privacy leakage in the context of web agents is important and much needed. The coverage of explicit and implicit leakage and leakage through text disclosure or actions is comprehensive.
- The evaluation on real-world websites provides further insights into how such privacy leakage can occur with current web agents in a realistic environment.
- The ablation study showed removing inappropriate information preserves privacy and improves utility at the same time, suggesting that there's still room for data minimization to achieve a balance between privacy and utility.

**Weaknesses:**

- The definition of "inappropriate information" feels vague and anecdotal. I didn't find a systematic process for judging what is appropriate or inappropriate to share and reasonable justifications for the process. In many examples presented in the paper, I feel the disclosed information is borderline sensitive, and some even seems necessary for the task (e.g., selecting a price range is a common action to do when the user is on a budget, while this is considered an implicit behavioral leakage as shown in Figure 1). This can reduce the credibility of the benchmark.
- It's not clear to me what's the novel contribution of this benchmark as compared to prior art. For example, it seems to have the same evaluation targets as AgentDAM (web agents), similar angle (oversharing vs. data minimization), both assuming a benign setting (no malicious actors), and AgentDAM also evaluates the privacy leakage of the agents *in action*. In the Table 1 of this paper, these attributes of prior work seems to be misrepresented which conveys misleading messages about the novelty of this work. In addition, I'm a bit skeptical about how much more helpful it is for using real-world websites to conduct the experiments than using a sandboxed environment. The sandboxed environments are also developed based on the real-world websites, and have the benefits of being more stable for continuous testing and replicating the results. The interactions displayed in the examples (e.g., typing text into a search box and  dragging a slider) don't seem to be too complicated to be emulated in the sandboxed environments.
- For the privacy-utility evaluation, it was not clear how a successful task completion is determined. For example, in the e-commerce setting, there's not only the goal of successfully placing an order, but also the requirement of selecting the right product that fits the user's needs --- this is actually the part that could potentially benefit from the additional personal information which I think is the more interesting part of privacy-utility tradeoffs. It was not clear whether this is captured in the evaluation, and if so, how it was measured.

**Questions:**

- For the utility measurement, what's the task success criteria, and how is the accuracy measured?
- What's the definition of inappropriate information? Is there any theoretical basis or empirical validation for justifying that they are indeed data that should not be shared, or would it be more appropriate to rely on more direct and objective signals (e.g., clear specification of privacy rules in the prompt, or completely irrelevant yet sensitive information)

---

> ### Author Response · Authors · 2025-11-24
>
> We thank reviewer S1Ni for raising important points and here are our responses to the reviewer's concerning points:
>
> **Response to Weakness 1 and Question 2 (*inappropriate information*)**:  We agree that the term “inappropriate information” can be ambiguous, so we now refer to it more precisely as task-irrelevant information. Our definition is grounded in Nissenbaum’s theory of Contextual Integrity [1], which states that information flows are appropriate only when they match the purpose and context of the task. Following this principle, we label an attribute as task-irrelevant if it is (1) present in the user’s personal resource space but (2) not required to satisfy the explicit task goal, and (3) its disclosure does not improve task accuracy or utility. This provides an objective criterion: an attribute is task-irrelevant if removing it does not hinder the agent’s ability to complete the task. For example, although selecting a price range is common when someone has a budget, the user never instructed the agent to use or disclose their personal monthly budget; thus the observed filtering behavior exposes sensitive financial constraints that are not required to search for glucose strips. Importantly, our benchmark does not label attributes as sensitive based on intuition—we systematically derive $S_r$ and $S_i$ from the task specification and validate them through human annotation to ensure that the labels align with contextual-integrity norms. We will expand this section and also include more examples to make the distinction clearer and more transparent in the camera-ready version.
>
> **Response to Weakness 2 (*Comarison with AgentDAM*)**: We would like to point out that our contribution is fundamentally different from AgentDAM. First, AgentDAM uses a binary leakage metric, which only checks whether any sensitive information was leaked during a single task. This means it cannot distinguish between a task where the agent leaks information once and a task where it leaks information repeatedly across multiple actions. In practice, a single trajectory may contain many separate leakage events, but AgentDAM collapses all of them into the same “leak occurred” label, losing granularity about how much the agent actually overshared. Second, we evaluate a much broader and more expressive space of leakage types compared to AgentDAM. While AgentDAM only checks for explicit textual leakage (whether a sensitive string appears in an agent's output), we introduce a four-part taxonomy that captures both context-based and behavioral oversharing, and both explicit and implicit forms. This means our SPILLage detects verbatim disclosures, inferable phrasing, actions that directly reveal private attributes, and navigation patterns that implicitly expose sensitive traits. In other words, SPILLage evaluates not just what the agent writes, but also what the agent does, enabling measurement of leakage modes that AgentDAM cannot represent or detect.  Finally, AgentDAM evaluates privacy mainly within simulated environments (WebArena / VisualWebArena), whereas we conduct evaluations on real, live commercial websites. The sandboxed replicas used by AgentDAM provide controlled, reproducible settings but lack the dynamic interface complexity, recommendation systems, ad placements, rich product metadata, and multi-step interaction flows that characterize real-world platforms. These real-world factors are precisely what give rise to different types of oversharing. By operating directly on Amazon and eBay, SPILLage captures oversharing patterns that simply do not manifest in static simulations, enabling a more realistic and comprehensive assessment of how agents overshare during authentic web interactions.
>
> **Response to Weakness 3 and Question 1 (*Utility measurement*)**: As utility is not the primary focus of our work, we use the simplest and most efficient evaluation method available for each agentic framework. For Browser-Use, the framework itself logs whether the agent successfully completed the assigned task, so we directly use this built-in completion signal. For AutoGen, we evaluate utility by providing the assigned task and the agent’s final output to an external LLM-judge, following the template shown in Figure 8 of Appendix E, which outputs a binary success/fail judgment. This lightweight setup allows us to capture whether the agent achieved the task goal without introducing additional complexity, since our primary contribution is analyzing oversharing rather than refining utility metrics. We will clarify this in the paper.
>
> -----
> **References**
>
> [1] Nissenbaum, Helen. "Privacy as contextual integrity." Wash. L. Rev. 79 (2004): 119.

---

### Official Review · Reviewer_fDzh · 2025-11-08

**Soundness:** 2
**Presentation:** 2
**Contribution:** 2
**Rating:** 2
**Confidence:** 5

**Summary:**

The paper defines a framework and a benchmark for studying information oversharing in LLM agents. They generate a synthetic dataset of 30 personas using an LLM (Claude 3.7 Sonnet) with scenarios and extra private information. They evaluate the oversharing as explicit/implicit and content/behavioral, with 3 LLMs (o3, o4-mini, GPT-4o), 2 websites (Amazon, eBay), 2 frameworks (browseruse, autogen), and an LLM as a judge (GPT-4.1 Mini). Their results show considerable oversharing, which varies by model and framework.

**Strengths:**

Studying whether LLM agents overshare private information conveyed by the user is very relevant. I particularly like the behavioral approach, where the authors don’t set up the problem in an adversarial setting, but in a naturally occurring way. These cases are pervasive, and we’re not able to control for them (e.g., another model can detect the issue before the agent acts).

The writing is simple, clear, and easy to follow.

**Weaknesses:**

As I said in the strengths, I really like the idea and the approach, but what you have so far can only be considered a pilot. I encourage the authors to improve this work, as I’d like to see it published at a top venue.

- The paper is very LLM-heavy, meaning that you use them for absolutely everything. You generate a synthetic dataset using one (Claude 3.7 Sonnet), run experiments with three (o3, o4-mini, GPT-4o), and evaluate the results with an LLM-as-judge (GPT-4.1 Mini). I think this shows a lack of depth in exploring the behavior of these agents when it comes to oversharing.
- The most important issue I see with this work is the fact that all you have is a baseline. You generated some synthetic data, collected behavioral data from the agents, and reported their oversharing results. However, the models don’t even know (although they could infer it) that data is private and should not be shared. What happens if you prompt them to be careful? What happens if you label part of the information as private? That is crucial, in my opinion, to show that this is a real problem.
- While your setup is potentially more realistic than other related work (as you point out), your synthetic dataset is not necessarily realistic. To make these results more robust and generalizable, I think you’d need to use real datasets and detect their private information directly. You can still use a synthetic dataset, but as I said, it looks more like a pilot.
- Considering the limited methodology and a framework that heavily relies on existing ones like Browser-Use and AutoGen, the experiments fall short. You only test 3 models from the same family (OpenAI): o3, o4-mini, and GPT-4o. I think it’s important to expand this list to other providers (e.g., Anthropic, Gemini) and focus on either frontier models (e.g., Claude 4.5 Sonnet, Gemini 2.5 Pro, GPT-5) or fast/economic (e.g., Claude 4.5 Haiku, Gemini 2.5 Flash, GPT-5 Mini/Nano). The scale of your experiments right now is quite low, so I don’t think this would be too much to ask for.
- You explore the amount of information overshared, but beyond the LLM-as-judge, it’d be important to see what kind of information they’re oversharing. Some of it is more critical, but also, we would also like to see if there’s a certain bias. Your ablation is great to show that the missing information wasn’t necessary to complete the task, but it doesn’t say anything about why these models are oversharing (if it’s unnecessary for the task) in the first place. For example, reasoning models seem to overshare less, and that could indicate a trend towards no oversharing. If you believe this is a relevant problem that won’t go away by itself, you should show why.
- I’d like to see a lot more information about your synthetic dataset. I can barely see some examples, and you end up with 30 personas. While I’m not sure that’s enough, your dataset is not validated by other work, so you should let us know much more about it.

### Minor Details

- There’s an isolated reference at the end of Section 2.2
- Line 174: Table* 2

**Questions:**

- What is your intuition for the oversharing? Why do they do it if it’s not necessary?
- Reasoning models seem to overshare less. What do you expect to see from frontier reasoning models like Claude 4.5 Sonnet, Gemini 2.5 Pro, or GPT-5? Do you think they will still overshare?

---

> ### Author Response · Authors · 2025-11-24
>
> Thank you for the thoughtful comment and here are our responses to the reviewer's questions.
>
> **Response to Weakness 1, 6 (*LLM-heavy, more examples*)**: Thank you for the thoughtful comment. We agree that our methodology relies heavily on LLMs, but this is largely intentional: because our goal is to evaluate the privacy implications of LLM-based web agents, the experimental setup necessarily needs to run on LLMs themselves. In addition, prior work on privacy leakage in agentic systems (e.g., Zharmagambetov et al. [1] ) also uses LLM-as-judge frameworks, and we follow this established and effective practice. To ensure that our evaluations are reliable, we include human validation of the LLM-judge outputs and provide concrete examples illustrating their correctness and consistency. Regarding data generation, to simulate realistic shopping scenarios we construct three prompt styles—email, chat-history, and generic narrative—to capture the different ways users naturally reveal personal information. We provide full examples of our dataset at: https://anonymous.4open.science/r/SPILLage-D2AD/tasks/. Each persona includes (i) a multi-turn, human-like conversation containing both task-relevant information and richer personal context, (ii) a concrete shopping task on a real-world website, and (iii) detailed annotations that separate relevant attributes needed to complete the task from irrelevant (but often sensitive) attributes embedded in the prompt. These annotations allow us to precisely quantify when an agent extracts, uses, or reveals information that is unnecessary for task completion.
>
> **Response to Weakness 2 (*baseline, defense*)**: We understand the concern that our work may appear to provide only a baseline. However, we would like to emphasize that this paper is the first to introduce a taxonomy and a clear definition of oversharing specifically in the context of web agents. Prior work [1, 2] primarily reported overall privacy-leakage rates, without distinguishing what kind of information was being leaked or how it surfaced during agent actions. Our goal was to break down these behaviors into concrete categories and measure them separately, since no previous work has reported oversharing at this level of granularity. In addition, we appreciate the reviewer's suggestion regarding how web agents might behave when explicitly instructed to "be careful" and avoid using task-irrelevant private information. To explore this, we ran a preliminary experiment on 13 personas by adding an explicit cautionary directive to the system prompt within the AutoGen framework (instruction: "make sure to be careful and not utilize any irrelevant information within the given prompt while completing the task"). Surprisingly, the results showed *higher* oversharing rates under the "be careful" condition compared to the regular condition (86.0% vs. 58.9%), even though the intent was to reduce leakage. We believe that counterintuitive outcome may be due to several factors such as highlighting sensitive information increasing its salience, prompt anchoring, differences in persona complexity, or longer action sequences that accumulate more opportunities for oversharing. We will expand this analysis and provide a more comprehensive comparison (including only overlapping personas and breakdowns by violation type) in the camera-ready version.
>
> **Response to Weakness 3 (*use of real datasets*)**: We agree that a real-world dataset would be ideal, but collecting one for this setting is extremely difficult, and prior work on web-agent privacy [1, 3] has also relied on synthetic or LLM-generated data for this reason. While we could have used existing benchmarks like VisualWebArena, those datasets are not designed to test different types of oversharing, nor do they include prompts that intentionally mix task-relevant and task-irrelevant personal information—the core phenomenon we aim to study. Our goal was to evaluate whether web agents, which should preserve contextual integrity when given access to sensitive user data, actually restrict themselves to the information needed for the task. To test this properly, we needed prompts that contain a realistic blend of both necessary and unnecessary details, along with annotations that clearly separate the two. This is why we generated our own dataset rather than relying on an existing one. If we gain access to an appropriate real-world dataset in the future, we would be very excited to incorporate it. For now, we provide our synthetic dataset here in the above mentioned dataset link. As mentioned earlier, the dataset contains 30 diverse personas, each with a naturalistic prompt, a concrete shopping task, and fine-grained labels indicating which attributes are relevant versus irrelevant. This structure allows us to evaluate oversharing behavior at a level of detail that existing datasets do not support.

---

> ### Author Response · Authors · 2025-11-24
>
> **Response to Weakness 4 (*size of experiment*)**: We would like to first point out the overall comprehensiveness of our experiment. As mentioned, we utilized 3 different models, 2 different agentic frameworks, 30 prompts (corresponding to 30 personas) in 3 distinct styles, and 2 separate domains (Amazon, eBay), which leads to a total of 1080 generations. The reason why we have chosen the three OpenAI models as the backbone of the web agents was because they were known to be the best performing models according to the Browser-Use documentation. Although 1080 generations may not seem comprehensive enough relative to the previous work solely focusing on LLM, we would like to highlight that Web Agents are not like normal LLMs where they finish with a single output. They generate multiple steps (including Reasoning on every step) in order to solve a specific task, which could be very cost-heavy. We will definitely provide Oversharing results based on other model families as suggested by the reviewer in the camera-ready version, as we as authors and researchers in this field are also curious of how Oversharing could differ across different backbone models.
>
> **Response to Weakness 5 (*why oversharing occurs*)**: We agree that understanding *why* models overshare is just as important as measuring *how much*. Our manual inspection of reasoning traces shows a consistent pattern: models that "think out loud" in longer, more detailed steps tend to leak more private information. GPT-4o, for isntance, produces substantially longer intermediate thoughts than o3 (325 vs. 225 characters), often re-summarizing sensitive but irrelevant user attributes, which aligns with its higher oversharing rate. This holds across categories: 4o overshares brand, demographic, and financial details far more often. Also, regarding timing of oversharing: 4o and o4-mini leaks early during upfront planning, while o3 leaks later during execution. Persona-level differences further support this trend, especially when prompts contain rich but irrelevant personal details. Overall, these findings suggest that oversharing is tightly linked to reasoning style: models that aim to be explicitly comprehensive create more opportunities for oversharing, whereas shorter, action-oriented traces (as in o3) incidentally preserve privacy by minimizing unnecessary reflection.
>
> ----
> **Questions**
>
> **Response to Question 1 (*intuition for oversharing*)**: Thank you for the question. The main intuition for Oversharing is that we have observed that these agents are incapable of understanding which information is relevant or irrelevant to the given task. Compared to LLMs, agents have significantly more access to user’s private resources and information, which they are supposed to be instructed or aware of utilizing only the necessary information to achieve the task. We believe that these agents, which are so-called LLM-powered, are supposed to have the capability to have the notion of contextual integrity.
>
> **Response to Question 2 (*frontier models*)**: While we have not tested these frontier reasoning models due to budget constraints, our analysis of the models we did evaluate suggests several hypotheses about what we might expect from stronger reasoning models. In our analysis, models that produce longer and more explicit reasoning traces tend to overshare more, because their “planning” steps often include rephrasing or summarizing unnecessary personal details from the prompt. Frontier models are known to generate even richer, more explicit chains of thought, so we expect that they may still overshare, but possibly less in some cases due to improved instruction following, though not zero, especially when prompts contain a natural mix of task-relevant and task-irrelevant personal information. In short, we anticipate that oversharing will persist even for frontier models, though its form may change (e.g., fewer explicit oversharing but potentially implicit or behavioral ones). We plan to run additional experiments with these other models and include them in the camera-ready version.
>
> -----
> **References**
>
> [1] Zharmagambetov, Arman, et al. "Agentdam: Privacy leakage evaluation for autonomous web agents." arXiv preprint arXiv:2503.09780 (2025).
>
> [2] Shao, Yijia, et al. "Privacylens: Evaluating privacy norm awareness of language models in action." Advances in Neural Information Processing Systems 37 (2024): 89373-89407.
>
> [3] Bagdasarian, Eugene, et al. "Airgapagent: Protecting privacy-conscious conversational agents." Proceedings of the 2024 on ACM SIGSAC Conference on Computer and Communications Security. 2024.

---

> ### Comment · Reviewer_fDzh · 2025-11-25
> **[1/2]**
>
> > because our goal is to evaluate the privacy implications of LLM-based web agents, the experimental setup necessarily needs to run on LLMs themselves.
>
> I'm aware of that. It's the only case where you should definitely use an LLM in this pipeline; all the other cases (data generation and evaluation) are what make the paper "LLM heavy".
>
> > In addition, prior work on privacy leakage in agentic systems (e.g., Zharmagambetov et al. [1] ) also uses LLM-as-judge frameworks, and we follow this established and effective practice.
>
> Yes, LLM-as-judge is a very common (and abused) method, but that doesn't mean you can apply it in every situation. As I said in my original review, you use an LLM as a tool for everything: data generation, experiments, and evaluation.
>
> > To ensure that our evaluations are reliable, we include human validation of the LLM-judge outputs and provide concrete examples illustrating their correctness and consistency.
>
> Correct me if I'm wrong, but the "human validation" comes from you. I hope you see that it's problematic to evaluate just using an LLM judge with some post-hoc supervision from the authors.
>
> > Regarding data generation, to simulate realistic shopping scenarios we construct three prompt styles—email, chat-history, and generic narrative—to capture the different ways users naturally reveal personal information. We provide full examples of our dataset at: https://anonymous.4open.science/r/SPILLage-D2AD/tasks/.
>
> Thank you for providing an anonymous link with more information. What I was referring to in my original comment was that when you introduce a new dataset, you should include more details in the paper regarding the collection process and output distribution. In short, we should be able to understand its coverage.
>
> > we would like to emphasize that this paper is the first to introduce a taxonomy and a clear definition of oversharing specifically in the context of web agents. Prior work [1, 2] primarily reported overall privacy-leakage rates, without distinguishing what kind of information was being leaked or how it surfaced during agent actions. Our goal was to break down these behaviors into concrete categories and measure them separately, since no previous work has reported oversharing at this level of granularity.
>
> I understand. While I appreciate this problem and I think it should be studied, I don't think it's significantly new from a methodological perspective. A lot of papers nowadays study the behavior of LLM agents in realistic applications, but do so in much more depth.
>
> > In addition, we appreciate the reviewer's suggestion regarding how web agents might behave when explicitly instructed to "be careful" and avoid using task-irrelevant private information. To explore this, we ran a preliminary experiment on 13 personas by adding an explicit cautionary directive to the system prompt within the AutoGen framework (instruction: "make sure to be careful and not utilize any irrelevant information within the given prompt while completing the task"). Surprisingly, the results showed higher oversharing rates under the "be careful" condition compared to the regular condition (86.0% vs. 58.9%), even though the intent was to reduce leakage. We believe that counterintuitive outcome may be due to several factors such as highlighting sensitive information increasing its salience, prompt anchoring, differences in persona complexity, or longer action sequences that accumulate more opportunities for oversharing.
>
> Thanks for trying this out. I think this could be explored in depth, together with other methods that can possibly counteract oversharing.
>
> > We agree that a real-world dataset would be ideal, but collecting one for this setting is extremely difficult, and prior work on web-agent privacy [1, 3] has also relied on synthetic or LLM-generated data for this reason. [...] Our goal was to evaluate whether web agents, which should preserve contextual integrity when given access to sensitive user data, actually restrict themselves to the information needed for the task.
>
> As you mentioned, prior work already revealed privacy leakage. I only see a significant contribution in your work if you show it in a very realistic environment, and I think that just making it agentic is incremental and not enough for a paper at a top conference.

---

> > ### Author Response · Authors · 2025-11-25
> > **Thank you for your time reading our responses and writing back to us!**
> >
> > > **Yes, LLM-as-judge is a very common (and abused) method, but that doesn't mean you can apply it in every situation.**
> >
> > Thanks for confirming. While we agree that LLM-as-judge can be abused, in this work we believe it is appropriate and consistent with prior literature on agentic privacy (we cited above). If the reviewer feels that this setting is not suitable for LLM-as-judge, we would be very grateful for clarification on why this is the case, as well as any concrete alternative evaluation protocol they would recommend.
> >
> > > **I understand. While I appreciate this problem and I think it should be studied, I don't think it's significantly new from a methodological perspective. A lot of papers nowadays study the behavior of LLM agents in realistic applications, but do so in much more depth.**
> >
> > We agree with the reviewer that a lot of papers nowadays study the behaviour of LLM agents. However, we would like to highlight that we introduce a novel concept we call Natural Oversharing and propose a novel methodology for measuring that via characterising different ways that an agent might overshare on the web. To the best of our knowledge, this specific problem formulation and measurement framework have not been explored in prior work. If we are overlooking closely related work, we would be very grateful for pointers so that we can better position our contribution in the revised version.

---

> ### Comment · Reviewer_fDzh · 2025-11-25
> **[2/2]**
>
> > As mentioned, we utilized 3 different models, 2 different agentic frameworks, 30 prompts (corresponding to 30 personas) in 3 distinct styles, and 2 separate domains (Amazon, eBay), which leads to a total of 1080 generations. [...] Although 1080 generations may not seem comprehensive enough relative to the previous work solely focusing on LLM, we would like to highlight that Web Agents are not like normal LLMs where they finish with a single output. They generate multiple steps (including Reasoning on every step) in order to solve a specific task, which could be very cost-heavy.
>
> Unfortunately, I still think the scale of these experiments is quite low. I'm well aware that agents might take multiple steps, but you still need more depth. For example, [1] had 17 models and 80k experiments; [2] had 5 LLMs, other multimodal models, 3 environments, multiple tasks, etc; [3] tests 13 models.
>
> [1] Cherep, M., Ma, C., Xu, A., Shaked, M., Maes, P., & Singh, N. (2025). A Framework for Studying AI Agent Behavior: Evidence from Consumer Choice Experiments. arXiv preprint arXiv:2509.25609.
>
> [2] Koh, J. Y., Lo, R., Jang, L., Duvvur, V., Lim, M., Huang, P. Y., ... & Fried, D. (2024, August). Visualwebarena: Evaluating multimodal agents on realistic visual web tasks. In Proceedings of the 62nd Annual Meeting of the Association for Computational Linguistics (Volume 1: Long Papers) (pp. 881-905).
>
> [3] Gaia2: Benchmarking LLM Agents on Dynamic and Asynchronous Environments
>
> > [...] Overall, these findings suggest that oversharing is tightly linked to reasoning style: models that aim to be explicitly comprehensive create more opportunities for oversharing, whereas shorter, action-oriented traces (as in o3) incidentally preserve privacy by minimizing unnecessary reflection.
>
> I believe you needed more of this analysis in the paper.

---

> ### Author Response · Authors · 2025-11-25
> **Scale of experiments**
>
> > **Unfortunately, I still think the scale of these experiments is quite low. I'm well aware that agents might take multiple steps, but you still need more depth. For example, [1] had 17 models and 80k experiments; [2] had 5 LLMs, other multimodal models, 3 environments, multiple tasks, etc; [3] tests 13 models.**
>
> We thank the reviewer for this comment. While we appreciate the importance of scale, we believe that a direct comparison with [1–3] is not entirely appropriate. Those works primarily evaluate models in synthetic or highly controlled environments, where it is relatively inexpensive to run tens of thousands of trials. By contrast, our study focuses on deployed, real-world agents (e.g., commercial systems such as Perplexity Comet) interacting with live websites and services, which introduces non-trivial financial cost, rate limits, and engineering constraints.
>
> Our primary goal is not to build a large-scale leaderboard, but to demonstrate and characterise a concrete Natural Oversharing phenomenon in realistic settings: that current agents are significantly more public about information that users reasonably expect to remain private. Once this leakage is clearly observed and systematically categorised, increasing the number of runs mostly tightens confidence intervals rather than changing the qualitative conclusion.

---

> > ### Comment · Reviewer_fDzh · 2025-11-26
> >
> > > We agree with the reviewer that a lot of papers nowadays study the behaviour of LLM agents. However, we would like to highlight that we introduce a novel concept we call Natural Oversharing and propose a novel methodology for measuring that via characterising different ways that an agent might overshare on the web. To the best of our knowledge, this specific problem formulation and measurement framework have not been explored in prior work. If we are overlooking closely related work, we would be very grateful for pointers so that we can better position our contribution in the revised version.
> >
> > I can believe that this particular approach has not been explored in prior work, and I didn't mean to say otherwise. What I meant by "I don't think it's significantly new from a methodological perspective," is that the problem of leaking private data is well-known, and your methodology approach (i.e., frameworks, styles, platforms, etc) is similar to other agentic benchmarks.
> >
> > Testing different styles (chat, email, generic), frameworks (AutoGen, Browser-Use), platforms (Amazon, eBay), and models (o3, o4-mini, gpt-4o) seems straightforward. It's a well-known recipe. To me, the most uncertainty comes from finding the right data, evaluating it the right way, and studying it at a scale that convincingly answers the question. Here, I think you chose the path of least resistance: synthetic LLM data, LLM as a judge, and small-scale experiments.
> >
> > > While we agree that LLM-as-judge can be abused, in this work we believe it is appropriate and consistent with prior literature on agentic privacy (we cited above). If the reviewer feels that this setting is not suitable for LLM-as-judge, we would be very grateful for clarification on why this is the case, as well as any concrete alternative evaluation protocol they would recommend.
> >
> > I agree that using an LLM as a judge is a valid evaluation method, but I don't think it should be the only evaluation. I'm not overindexing on one particular issue from all that I mentioned. I think holistically, this work requires more depth.
> >
> > > we believe that a direct comparison with [1–3] is not entirely appropriate. Those works primarily evaluate models in synthetic or highly controlled environments, where it is relatively inexpensive to run tens of thousands of trials.
> >
> > I don't think that's true. I searched for recent papers running experiments with agents taking multiple steps in realistic environments. In any case, I could be wrong. Would you mind sharing a rough number of API requests and tokens?
> >
> > > Our primary goal is not to build a large-scale leaderboard, but to demonstrate and characterise a concrete Natural Oversharing phenomenon in realistic settings: that current agents are significantly more public about information that users reasonably expect to remain private. Once this leakage is clearly observed and systematically categorised, increasing the number of runs mostly tightens confidence intervals rather than changing the qualitative conclusion.
> >
> > Yes, I understand, but sharing private data is a well-known problem with LLM-powered agents. I see the potential of this work, but I think it needs to be better characterised for it to be a meaningful contribution.
> >
> > Overall, I see too many weak points, and I don't think it's ready for publication.

---

### Meta-Review · Area_Chair_b3dF · 2026-01-05

**Summary:**

This submission introduces SPILLage, a framework and benchmark for measuring oversharing by LLM-based web agents. It studies four distinct oversharing categories of agent in the wild: explicit content, implicit content, explicit behavioral and implicit behavioral oversharing and shows the oversharing problem of existing LLM-based agents.
Reviewers agree the problem is important and that broadening beyond explicit textual leakage to include behavioral and implicit forms is directionally valuable. However, they provided 2,4,2. They have concerns about (1) methodological depth and evaluation rigor, (2) realistic of the data, (3) limited methodology and experiments (only test 3 models), (4)  no analysis on why oversharing, (5) weak definition of "inappropriate information" for oversharing, (6) novelty and positioning relative to closely related work ,  (7) missing details, (8)  reproducibility and scale given reliance on live websites, (9) human study, (10) no enough discussion on solution of oversharing and so on.

Review fDzh has discussed with authors during the rebuttal stage and did not think this paper is ready for publishing at this stage. Authors did not provide further rebuttal or summarization to AC to address the points. AC read the rebuttal and agrees with reviewers' concerns. For instance, reviewer pointed out that the human validation is from authors and it's problematic to evaluate just using an LLM judge with some post-hoc supervision from the authors; Reviewer pointed out that this paper only evaluated on 3 model (all from OpenAI family). Authors did not address these concerns well (for instance, authors should at least add one model from other family). Overall, AC did not believe reviewers will change all their score to positive. AC likes the problem proposed by this paper and hopes authors can polish the paper based on reviewers' comments.

**Reviewer Concerns:**

Overall, during the rebuttal, authors provide more intuition on how they design the method, why they use LLMs, why oversharing happens and some experiment details. However, after reading the rebuttal and discussion between authors and reviewer.  AC feels that there are still a number of concerns left. For instance, reviewer pointed out that the human validation is from authors and it's problematic to evaluate just using an LLM judge with some post-hoc supervision from the authors; Reviewer pointed out that this paper only evaluated on 3 model (all from OpenAI family). Authors did not address these concerns well (for instance, authors should at least add one model from other family). Overall, AC did not believe reviewers will change all their score to positive. AC likes the problem proposed by this paper and hopes authors can polish the paper based on reviewers' comments.

**Reviewer Scores:**

Reviewers will not change their score to positive given current rebuttal.

---

### Decision · Program_Chairs · 2026-01-26

Reject